# Using an on-site modular training approach to amplify prep service delivery in public health facilities in Kenya

Elizabeth M. Irungu[1,2]*, Moses Musau[1], Bernard Nyerere[3], Anabelle Dollah[3], Benn Kwach[3], Emmah Owidi[1], Elizabeth Wamoni[1], Josephine Odoyo[3], Nelly Mugo[1,2], Elizabeth Bukusi[2,3,4], Kenneth Mugwanya[2], Jared M. Baeten[2,5,6¤], for the Partners Scale-Up Project Team

1 Centre for Clinical Research, Kenya Medical Research Institute, Nairobi, Kenya, 2 Department of Global Health, University of Washington, Seattle, WA, United States of America, 3 Centre for Microbiology Research, Kenya Medical Research Institute, Nairobi, Kenya, 4 Obstetrics and Gynecology, University of Washington, Seattle, WA, United States of America, 5 Epidemiology, University of Washington, Seattle, WA, United States of America, 6 Medicine, University of Washington, Seattle, WA, United States of America

¤ Current address: Gilead Sciences, Foster City, California, United States of America
* eirungu@uw.edu

**Data Availability Statement:** The data from the Partners Scale-Up Project are available by contacting the International Clinical Research Center at the University of Washington (icrc@uw.

## Abstract

Delivery of oral PrEP, a potent HIV prevention intervention, has begun within public health systems in many countries in Africa. Training as many health providers as possible expeditiously is necessary to efficiently and rapidly scale up PrEP delivery among at risk populations and thereby realize the greatest impact of PrEP. We designed and implemented an innovative on-site modular training approach delivered in five two-hour modules. The modules could be covered in two consecutive days or be broken across several days enabling flexibility to accommodate health provider work schedules. We assessed knowledge gain comparing pre-and post-training test scores and determined monthly PrEP uptake for six months following the training intervention. We also evaluated the cost of this training approach and conducted key informant interviews to explore acceptability among health providers. Between January 2019 and December 2020, 2111 health providers from 104 health facilities were trained on PrEP. Of 1821 (83%) providers who completed both pre- and post-tests, 505 (28%) were nurses, 333 (18%) were HIV counsellors, 276 (15%) were clinical officers and 255 (14%) were lay providers. The mean score prior to and after training was 58% and 82% respectively (p <0.001). On average, health facilities initiated an average of 2.7 (SD 4.7) people on PrEP each month after the training, a number that did not decline over six months post-training (p = 0.62). Assuming Ministry of Health costs, the costs per provider trained was $16.27. Health providers expressed satisfaction with this training approach because it enabled many providers within a facility receive training. On-site modular training is an effective approach for improving PrEP education for health workers in public health facilities, It is also acceptable and low-cost. This method of training can be scaled up to rapidly amplify the number of health workers able to offer PrEP services.

edu). The authors have not shared the data publicly for ethical reasons and the Kenyan IRB (KEMRI-SERU) requires review of proposals for data access. The contact information for is KEMRI-SERU: Secretary, KEMRI Scientific and Ethics Review Unit (SERU), P.O. Box 54840-00200, Nairobi. Telephone numbers: 0202722541,0722205901, 0717719477 E-mail address: seru@kemri.org.

**Funding:** The Partners Scale-Up Project is supported by the National Institute of Mental Health of the US National Institutes of Health under grant R01 MH095507 and the Bill & Melinda Gates Foundation under grant OPP1056051 awarded to JMB. The funders had no role in study design, data collection and analysis, decision to publish, or preparation of the manuscript.

**Competing interests:** The authors declare they have no competing interests.

## Introduction

Of the estimated 38 million people living with HIV globally, more than 25 million live in sub-Sahara Africa, making it a region disproportionately burdened with HIV [1]. In 2019, close to 60% of the estimated 1.7 million new HIV infections were from the region and over 400,000 people died from the disease. In September 2015, the World Health Organization (WHO) recommended pre-exposure prophylaxis (PrEP) using oral daily medication tenofovir disoproxil fumarate (TDF), in combination with emtricitabine (FTC/TDF), for HIV prevention among people at risk of acquiring HIV [2,3]. Successful implementation of PrEP in the region is needed in order to attain impactful coverage and meet HIV-prevention goals.

Many countries in Africa have instituted PrEP programs, but scale up has been sub-optimal [4–6]. Early challenges observed with PrEP introduction include low community awareness and knowledge, service delivery points that are over-burdened and resource-constrained and may not accessible by some high risk populations, and stigma and discrimination associated with being at risk for HIV [4]. Improving providers PrEP education is essential to foster PrEP implementation and scale up [7,8], as trained health workers are more willing to discuss PrEP services with their clients and in the community [9–11]. In addition, health workers who are knowledgeable about PrEP and are competent in PrEP service delivery will identify individuals at risk of HIV acquisition, initiate them on PrEP and conduct follow up visits [7].

The national roll out of the Kenyan PrEP program began in 2017 [12]. Over 90,000 people have initiated PrEP in the country making it the second largest program in the continent [13]. However, only a limited number of health care providers in Kenya are trained on PrEP service delivery creating a barrier to expanding PrEP service delivery in public health facilities [14,15]. Training as many health providers as possible expeditiously is necessary to rapidly scale up PrEP delivery among at risk populations and thereby realize the greatest impact of PrEP in the country.

Here, we describe the implementation of an innovative on-site, modular PrEP training in public health facilities to amplify the number of providers knowledgeable and competent to offer PrEP services in Kenya. We evaluated the effectiveness and cost of this approach and explored acceptability among health providers.

## Methods

The Partners Scale-Up Project is an implementation science project that aims to catalyze national scale up of PrEP in public HIV care clinics in central and western regions in Kenya (Clinicaltrials.gov NCT03052010) [16]. The project initially focused on 25 high-volume clinics, then expanded to include training and providing technical assistance to health workers in additional facilities. The present report focuses on trainings developed within the first 25 clinics then implemented in the additional clinics.

Within this project we implemented on-site modular PrEP training sessions in public health facilities in Kenya. We first developed training modules i.e., self-contained sections that when combined with other sections constitute the entire PrEP training curriculum (*S1 Text*). These were modified from the national PrEP training curriculum that has been described elsewhere [12,14]. In brief, the curriculum had content on determination of PrEP eligibility, tasks to be performed at PrEP initiation and follow-up visits, commodity management and monitoring and evaluation (Table 1). The modules could be covered in two consecutive days or be broken across several days enabling flexibility to accommodate health provider work schedules.

The county HIV/AIDS leadership, including health facility management in collaboration with technical assistants from the project identified health facilities that would receive training.

**Table 1. Modular structure of the PrEP curriculum.**

| | Title | Description of Content |
|---|---|---|
| **Module 1** | Introduction to PrEP | • Provides a background of HIV burden in Kenya<br>• Describes what PrEP is and details PrEP efficacy<br>• Delineates who is eligible for PrEP<br>• Differentiates PrEP from PEP |
| **Module 2** | The Service Provider Toolkit | • Guides providers how to conduct risk assessment<br>• Provides guidance for how to initiate PrEP and follow up PrEP users |
| **Module 3** | Clinical Case Management | • Allows providers to role play and discuss various case scenarios |
| **Module 4** | PrEP Commodity Management | • Details flow of PrEP commodities and ordering processes.<br>• Guides providers how to do pharmacovigilance for PrEP |
| **Module 5** | Monitoring and Evaluation for PrEP | • Introduces providers to various reporting tools for PrEP |

Health facility management then determined the providers who would be trained, identified the training venue within the health facility where the training sessions would be conducted and proposed training dates that were convenient for them. Health providers working in the facility who were likely to come into contact with persons at risk of acquiring HIV or likely to be involved in provision of PrEP services were eligible for training. On-site training within the health facility allowed as many providers as were available to participate. The training sessions were co-facilitated by project staff and PrEP trainers from the county, if available. Each participant received a copy of the participants' training manual (*S2 Text*), the PrEP toolkit for providers (i.e., a manual of procedures for providing PrEP services) (*S3 Text*), and stationery. The training manual and the PrEP toolkit were previously developed by the Kenya National AIDS and STI Control Program (NASCOP) in collaboration with implementing partners, who are members of the national PrEP Technical Working Group (TWG). Interactive learning methods including the traditional didactic approach, clinical case discussions, and role-plays were applied throughout the in-person training. The duration for each module was two hours. Refreshments were offered during the training, but transport reimbursement was not provided.

## Data collection and analysis

**Knowledge gain and PrEP uptake.** A pre-test assessment was administered to the trainees prior to the start of the training (*S4 Text*). The same tool was administered at the end of the last module in order to assess training effectiveness in terms of knowledge gain following exposure to training content. This assessment was the same as the one developed for the standard national training curriculum. Providers were considered to have completed the training if they completed both pre-and post-test assessments. We computed the mean scores for the pre- and post-tests and using a paired t-test, compared mean pre-and post-test scores among providers who completed the training.

Using programmatic data obtained from health facilities at the beginning of training and for 6 months thereafter we obtained the number of individuals initiating PrEP per facility and ascertained the mean monthly PrEP uptake across health facilities following the modular training. To determine whether the number of monthly PrEP initiations per clinic changed over the six-month period after modular training, we conducted an analysis using a negative binomial mixed effects model with log link. The model included month since training and training

year as fixed effects and a random effect for each health facility. Analyses were conducted using R software 3.5.2 and Stata version 15 (StataCorp, College Station, TX).

**Cost analysis.** In fourteen randomly selected health facility trainings, we conducted activity based micro-costing following established guidelines to determine the programmatic cost per provider trained assuming implementation by the Kenyan Ministry of Health [17]. We obtained costs of training implementation from project expense reports and receipts. We included costs of training materials (including training manuals and stationery), refreshments, and training facilitator time. We obtained health provider salaries from published Kenyan civil service salary scales [18]. We excluded project related costs that would not be applicable in programmatic roll out such as travel and accommodation costs for project staff trainers and replaced project staff salaries with public sector salaries. We did not include costs of developing the training curriculum since the existing curriculum could be scaled to additional facilities as-is.

We computed the average cost of a modular training as the sum of training costs of the randomly selected trainings divided by the number of trainings. The cost per provider trained was then calculated as the average cost of modular training divided by the average number of providers per training. Expenditures reported in Kenyan shillings were converted to United States dollars (USD) [19]. Analyses were conducted in Excel (version 16.5, Microsoft, Redmond, WA).

**Qualitative interviews.** Using a semi-structured guide, we conducted in-depth interviews with 35 health providers who attended modular training sessions to gain a deeper understanding of the training experience and acceptability of the modular training approach (*S5 Text*). We purposively sampled health providers of varied cadres in different health facilities to capture different perspectives. Interviews were conducted in English face-to-face or via telephone and recorded. They were then transcribed verbatim. Transcripts were analyzed in Dedoose (Sociocultural Research Consultants LLC, Los Angeles, CA). An initial codebook was developed deductively from the interview guide. Additional codes were added inductively as initial transcripts were reviewed and coded. The first three transcripts were double coded by at least two members of the study team and inconsistent results were reviewed by the coders until consensus was reached. The remaining transcripts were coded independently by one member of the study team and reviewed by another member. After all data were coded, investigators used an iterative process of reading transcripts, comparing and contrasting coding, and identifying convergent and divergent themes within and between transcripts.

### Ethical review

Ethical approval for this project was obtained from the University of Washington Human Subjects Division and the Kenya Medical Research Institute Scientific Ethical Review Unit. All health workers participating in-depth interviews provided written informed consent if the interview was conducted in person or verbal consent if the interview was conducted via telephone; quantitative program data was deemed part of quality assessment and thus not requiring individual consent.

## Results

### Knowledge gain and PrEP uptake

Between January 2019 and December 2020, 2111 health providers from 104 health facilities received on site PrEP training using the modular training approach. The median number of participants per training was 19 (inter-quartile range 14–25). Of 1821 (83%) providers who completed both pre- and post-test assessments, 505 (28%) were nurses, 333 (18%) were HIV

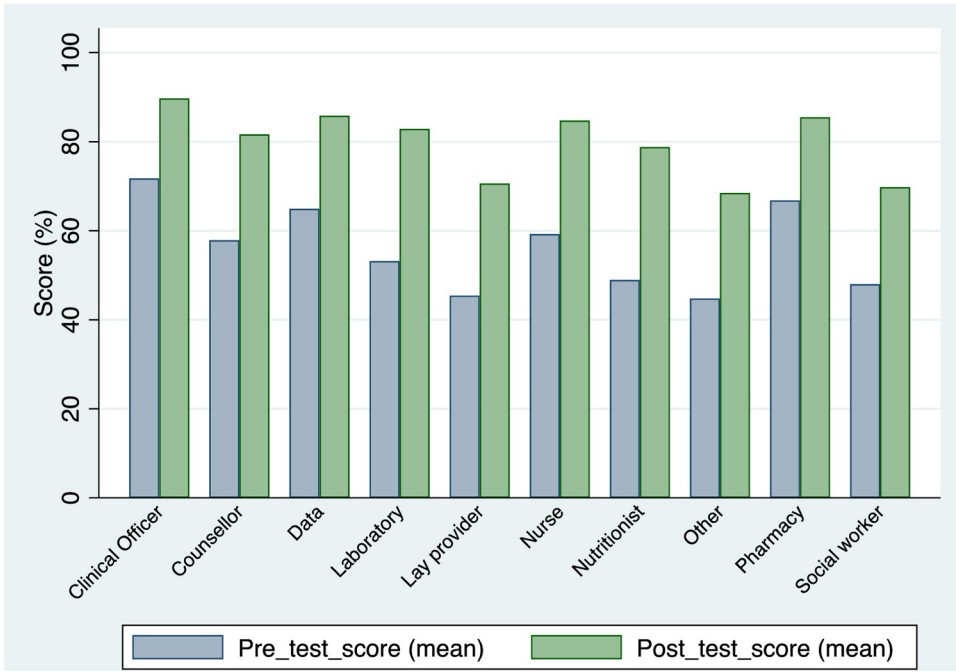

**Fig 1. Pre- and post-test mean scores, by cadre.**

counsellors, 276 (15%) were clinical officers and 255 (14%) were lay providers. The mean score prior to and after training was 58% and 82% respectively (p<0.001). Mean scores for all cadres increased significantly (p<0.001) after training compared to before training (Fig 1).

Over a six month follow up period post training, health facilities initiated an average of 2.7 (SD 4.7) people on PrEP per month, and this rate of PrEP initiations was consistent over the six months (p = 0.62).

## Cost of modular training

The overall project cost of implementation of on-site PrEP modular training across the 14 randomly selected facilities was $387 per facility training. The expected programmatic costs of implementing on-site modular training, assuming Ministry of Health costs is $315 per training and $16.27 per provider trained. Training materials, refreshments and trainer facilitation fees contributed 38%, 32% and 30% of the costs, respectively (Fig 2).

## Acceptability of modular training

Health providers reported that they gained knowledge from the training and felt competent enough to inform their clients about PrEP and offer PrEP services. They liked this training approach because it enabled many staff members, including key front-line providers and those working in various departments receive PrEP training. On-site training was unlike hotel-based trainings where only a few staff members are selected to participate. Providers reported that the advantage of having many staff members in the facility trained was that they would be comfortable talking to potential PrEP clients wherever they were stationed. They also stated that training together as peers facilitated interactive training sessions with open discussions that addressed facility-specific issues of PrEP delivery. In addition, health providers reported

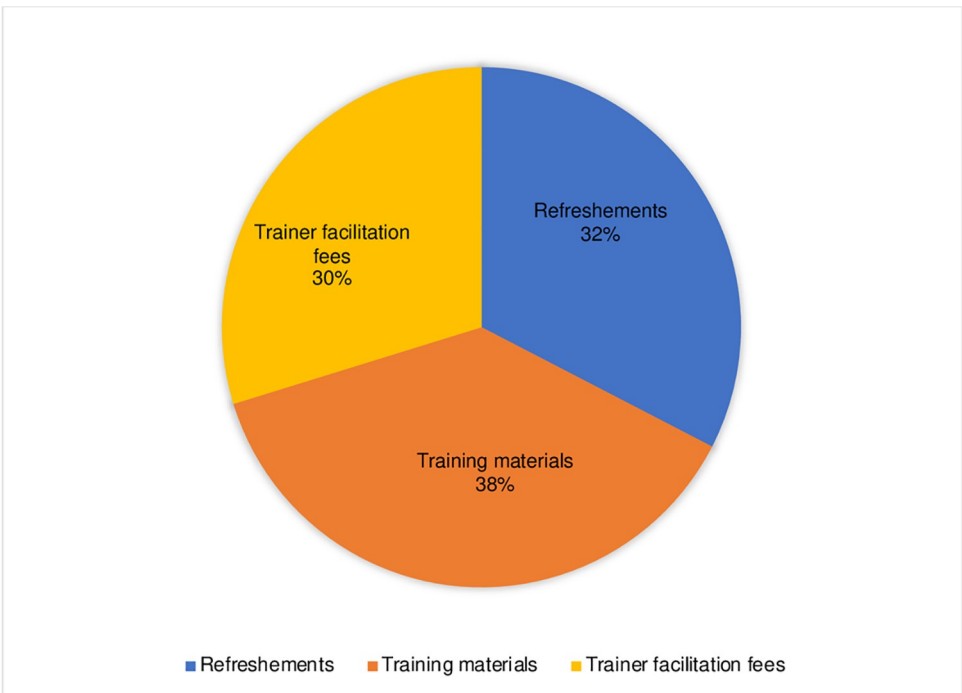

**Fig 2. Proportion of costs related to each resource type of on-site modular PrEP training for health providers.**

that training within the facility avoided the inconvenience of travelling out of facility for training.

> You know at times, you might find someone going for a training but they don't offer the service, they are not the key providers for the service. So, the facility-based training is like an advantage whereby whoever is implementing gets the chance to [get] first-hand information. . .. Then, an added advantage is even the rest get to know [about PrEP delivery]. (KII, Clinical Officer))

> There was a lot of discussion . . . towards how we can make the delivery better and make our facility also shine in terms of making sure that we have more and more people who are affected by the situation [HIV risk] come and get into PrEP. (KII, Nurse)

However, providers reported disruptions during the training in order to attend to patients, when workload increased. They also stated that they would have wished to receive some money, akin to transport reimbursement provided in out-of-facility trainings.

## Discussion

In this evaluation of an innovative on-site modular PrEP training curriculum, we trained over 2,000 health workers over a two-year period and found that health providers of all cadres gained PrEP knowledge following delivery of the curriculum. In addition, participating health facilities initiated at risk persons on PrEP at a rate that did not decline over the first six months after the training.

As PrEP services are largely offered in public health facilities, training all cadres of health providers on PrEP service delivery is paramount [6,20]. Trained front-line providers help their clients reflect on their HIV risk, they determine their eligibility for PrEP and issue

prescriptions to those eligible. As described in other settings, lay providers, when trained, are able to deliver health talks and have one-on-one conversations to educate people about PrEP [21,22]. Providers not primarily involved in PrEP delivery were also included in the training. This was important so that they could identify at-risk individuals from any department in the health facility and refer them to where they could get services. Importantly, training all people working in a health facility is valuable as the community looks to anyone working in health facilities, regardless of their cadre, to clarify PrEP-related information and help dispel myths related to PrEP [9].

Capacity-building efforts for health workers may be expensive and low-cost PrEP training modalities are urgently needed [23–27]. By offering training within health facilities, we did not incur costs of conferencing facilities and transport reimbursement. The on-site modular training approach was thus relatively inexpensive at $16 per provider trained. Other innovative cost-effective training approaches have been utilized including use of web-based training modules and on-job training [28,29].

The qualitative assessment found that the training impacted health providers' ability to offer PrEP services positively. The on-site modular training approach was acceptable, allowed flexibility with work schedules and in this way enabled many health providers to participate. In addition, the ability to be trained together as members of the same facility provided an opportunity for the participants to discuss PrEP delivery challenges unique to their facility and explore how to overcome them. Similar to our study, other studies have documented the feasibility and positive impact of on-site HIV-related training approaches [30,31].

There are limitations to these analyses. We did not have information on monthly PrEP initiation per facility prior to the training, and thus could not assess whether the PrEP initiation trend we observed post training was different from what had been happening in facilities prior to the training. The participating facilities were from central and western regions of Kenya, nevertheless they reflect the spectrum of public health facilities in Kenya, making the lessons learned generalizable to the rest of the country.

Optimizing access to PrEP requires expanding the number of healthcare providers who are knowledgeable and competent to deliver PrEP integrated in routine practice rapidly. On site modular training is an effective way to provide PrEP education for health workers in public health facilities. It is also an acceptable and low-cost approach. This method of training can be scaled up to rapidly amplify the number of health workers able to offer PrEP services.

## Supporting information

**S1 Text. Slides used during the facility-based modular training.**
(PDF)

**S2 Text. Participants' manual: A training manual that contains various case scenarios and role plays.**
(PDF)

**S3 Text. Service Provider Toolkit: A document that has additional detailed information to guide health providers offer PrEP services.**
(PDF)

**S4 Text. Pre- and post-test assessment: This tool was administered before and after training.**
(PDF)

**S5 Text. Topic guide for interviews with health providers.**
(PDF)

**S6 Text. List of modular training sites and training dates.**
(PDF)

## Acknowledgments

We are grateful to the management and health providers in all participating HIV care clinics.

The Partners Scale-Up Project trainers:

Western team: Josephine Odoyo, Bernard Nyerere, Kenneth Odhiambo, Meresa Oyier, John Bosco Tsetso

Central team: Elizabeth Wamoni, Margaret Mwangi, Roy Njiru, Irene Wanyoike, Winnie Waituika

## Author Contributions

**Conceptualization:** Nelly Mugo, Elizabeth Bukusi, Kenneth Mugwanya, Jared M. Baeten.

**Data curation:** Elizabeth M. Irungu, Moses Musau, Bernard Nyerere.

**Formal analysis:** Elizabeth M. Irungu, Moses Musau, Anabelle Dollah, Benn Kwach, Emmah Owidi, Kenneth Mugwanya.

**Funding acquisition:** Jared M. Baeten.

**Methodology:** Bernard Nyerere, Anabelle Dollah, Benn Kwach, Emmah Owidi, Elizabeth Wamoni, Josephine Odoyo, Kenneth Mugwanya.

**Project administration:** Elizabeth M. Irungu, Elizabeth Wamoni, Josephine Odoyo.

**Supervision:** Nelly Mugo, Elizabeth Bukusi, Kenneth Mugwanya, Jared M. Baeten.

**Writing – original draft:** Elizabeth M. Irungu.

**Writing – review & editing:** Elizabeth Wamoni, Josephine Odoyo, Nelly Mugo, Elizabeth Bukusi, Kenneth Mugwanya, Jared M. Baeten.

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
