## [Decision Letter · Decision Letter 0]

28 Sep 2021

PGPH-D-21-00431

USING AN ON-SITE MODULAR TRAINING APPROACH TO AMPLIFY PREP SERVICE DELIVERY IN PUBLIC HEALTH FACILITIES IN KENYA

Dear Ms. Elizabeth M. Irungu,

Thank you for submitting your manuscript to PLOS Global Public Health. After careful consideration, we feel that it has merit but does not fully meet PLOS Global Public Health’s publication criteria as it currently stands. Therefore, we invite you to submit a revised version of the manuscript that addresses the points raised during the review process.

We look forward to receiving your revised manuscript.

Kind regards,

The Academic Editor

Journal Requirements:

1. Please provide the dates of on-site training and a list of facilities where training took place.

2. In your Methods section, please provide additional information about the training participants. Please ensure you have provided sufficient details to replicate the analyses such as: 

a) the recruitment date range (month and year), 

b) a description of any inclusion/exclusion criteria that were applied to training participant recruitment, 

c) a table of relevant demographic details, 

d) a statement as to whether your sample can be considered representative of a larger population.

3. Please provide additional information on the source of supplementary file titled 'PrEP Toolkit for Providers' and 'Participants Manual'.

4. We have noticed that you have uploaded supporting information but you have not included a list of legends.  Please add a full list of legends for all supporting information files (including figures, table and data files) after the references list. 

5. Since your data is not available for proprietary reasons, please explain via email why the data is not available. Please also include the contact information for the third party organization that should be contacted should other researchers want to request access to this data and please include the full citation of where the data can be found. We also request that you verify with us via email that any researcher will be able to obtain the data set in the same manner that the you have obtained it. If you feel you are unwilling or unable to adhere to this policy, please explain your reasons by return email and your exemption request will be escalated to the editor for approval. Your exemption request will be handled independently and will not hold up the peer review process, but will need to be resolved should your manuscript be accepted for publication. One of the Editorial team will be in touch if they require more information.

Additional Editor Comments (if provided):

Reviewers' comments:

Reviewer's Responses to Questions

**Comments to the Author**

1. Does this manuscript meet PLOS Global Public Health’s publication criteria? Is the manuscript technically sound, and do the data support the conclusions? The manuscript must describe methodologically and ethically rigorous research with conclusions that are appropriately drawn based on the data presented.

Reviewer #1: Yes

Reviewer #2: Yes

2. Has the statistical analysis been performed appropriately and rigorously?

Reviewer #1: Yes

Reviewer #2: Yes

3. Have the authors made all data underlying the findings in their manuscript fully available (please refer to the Data Availability Statement at the start of the manuscript PDF file)?

Reviewer #1: Yes

Reviewer #2: Yes

4. Is the manuscript presented in an intelligible fashion and written in standard English?

Reviewer #1: Yes

Reviewer #2: Yes

5. Review Comments to the Author

Reviewer #1: This is a well written manuscript describing the impact and cost of an on-site multi-component provider training module for oral PrEP provision in Kenya using a before-after design. It underscores the importance of providers as gate-keepers to access to interventions and the importance of these gatekeepers to be well trained and confident to facilitate access to the new technology being introduced. It does require a potential user to arrive at a health facility in the first instance and strategies to enhance uptake need to be reviewed regularly as uptake and coverage increases and the intervention becomes normative. Two pints for consideration: i. beyond the basics the needs of potential users vary quite a bit and it would have been great eg to have the authors include some inputs on for example strategies on reaching adolescent girls and young women or men through this type of training. ii. in future 'before-after' impact assessments it may be useful to consider including a group that was not exposed to the 'pre-'/before assessment as this may help mitigate a bias inherent in before-after impact assessments.

Reviewer #2: 1. Introduction - general comment: The authors chose to study providers modular training as away to improve PrEP uptake/utilization. This is premised on the fact that inadequate training is one of the barrier to PrEP utilization/uptake, right? The authors should consider setting the stage by discussing barriers to PrEP uptake/use to give context to the role played by provider education- in other words, strengthen on the why the provider education and the choice of the training re; modular

2. Introduction, line 59: the authors describe their approach as innovative. What make this innovative (over other approaches)? Never tried before?

3. Conclusion line 27 and 253: The authors conclude their modular training is effective. In the absence of data (e.g. how many people were being initiated on PrEP pre-training - which is acknowledged in the limitation), and how "effectiveness" was defined in the study, it is difficult to verify that the modular training was effective. The authors should consider reviewing this by providing more context or data to support the effectiveness of the training or tone down on the language

4. Line 82: Health facility management determined the providers to be trained? According to a specific guidance? If there was no specific criteria guiding facility managers on how to pick the providers, there is a likely variation in cadres of staff picked by each facility and/or introduce selection bias? This has been pointed as a frequent occurrence in previous trainings as pointed by one of the participant (line 197) - the participant does allude this did not occur in their facility but I think the authors need to be clear on how they manage it

6. PLOS authors have the option to publish the peer review history of their article (what does this mean?). If published, this will include your full peer review and any attached files.

**Do you want your identity to be public for this peer review?** For information about this choice, including consent withdrawal, please see our Privacy Policy.

Reviewer #1: No

Reviewer #2: No

---

## [Editor Report · Decision Letter 1]

22 Dec 2021

USING AN ON-SITE MODULAR TRAINING APPROACH TO AMPLIFY PREP SERVICE DELIVERY IN PUBLIC HEALTH FACILITIES IN KENYA

PGPH-D-21-00431R1

Dear Dr. Irungu,

We're pleased to inform you that your manuscript has been judged scientifically suitable for publication and will be formally accepted for publication once it meets all outstanding technical requirements.

Within one week, you'll receive an e-mail detailing the required amendments. When these have been addressed, you'll receive a formal acceptance letter and your manuscript will be scheduled for publication.

An invoice for payment will follow shortly after the formal acceptance. To ensure an efficient process, please log into Editorial Manager at https://www.editorialmanager.com/pgph/ click the 'Update My Information' link at the top of the page, and double check that your user information is up-to-date. If you have any billing related questions, please contact our Author Billing department directly at authorbilling@plos.org.

Kind regards,

Samiratou Ouédraogo, DPharm, MPH, Ph.D.

The Academic Editor